# Novel Therapies for Unmet Clinical Needs in Myelodysplastic Syndromes

**DOI:** 10.3390/cancers14194941

**Published:** 2022-10-09

**Authors:** Giulio Cassanello, Raffaella Pasquale, Wilma Barcellini, Bruno Fattizzo

**Affiliations:** 1Department of Oncology and Oncohematology, University of Milan, 20122 Milan, Italy; 2Hematology Unit, Fondazione IRCCS Ca’ Granda Ospedale Maggiore Policlinico, 20122 Milan, Italy

**Keywords:** myelodysplastic syndromes, unmet needs, target therapy, somatic mutations, immunotherapy

## Abstract

**Simple Summary:**

Several novel therapies are being developed to improve the management of patients with myelodysplastic syndromes. They include drugs aimed at improving hematopoiesis and differentiation of myeloid precursors, hypomethylating agents, several compounds that target intracellular molecular pathways, and immunotherapies. In this review article, we discuss how the novel drugs may address the several unmet needs of lower- and higher-risk patients.

**Abstract:**

Myelodysplastic syndromes (MDS) are a very heterogeneous disease, with extremely variable clinical features and outcomes. Current management relies on risk stratification based on IPSS and IPSS-R, which categorizes patients into low (LR-) and high-risk (HR-) MDS. Therapeutic strategies in LR-MDS patients mainly consist of erythropoiesis stimulating agents (ESAs), transfusion support, and luspatercept or lenalidomide for selected patients. Current unmet needs include the limited options available after treatment failure, and the consequent transfusion burden with several hospital admissions and poor quality of life. Therapeutic approaches in HR-MDS patients are aimed at changing the natural course of the disease and hypometylating agents (HMA) are the first choice. The only potentially curative treatment is allogeneic stem cell transplant (allo-HCT), restricted to a minority of young and fit candidates. Patients unfit for or those that relapse after the abovementioned options harbor an adverse prognosis, with limited overall survival and frequent leukemic evolution. Recent advances in genetic mutations and intracellular pathways that are relevant for MDS pathogenesis are improving disease risk stratification and highlighting therapeutic targets addressed by novel agents. Several drugs are under evaluation for LR and HR patients, which differ by their mechanism of action, reported efficacy, and phase of development. This review analyzes the current unmet clinical needs for MDS patients and provides a critical overview of the novel agents under development in this setting.

## 1. Introduction

Myelodysplastic syndromes (MDS) are a heterogeneous group of diseases characterized by clonal hematopoiesis and abnormal cellular maturation, leading to one or more peripheral blood cytopenias (i.e., anemia, neutropenia, and/or thrombocytopenia) [1]. The pathophysiology consists of a multi-step process that involves cytogenetic changes and/or gene mutations [2], gene hypermethylation, and abnormalities of the bone marrow microenvironment [3,4]. The median age of onset is ~70 years and only a small minority of younger and fit patients may benefit from acute myeloid leukemia (AML)-like chemotherapy and be potentially cured by allogeneic hematopoietic stem cell transplant (allo-HCT) [5]. For the vast majority of subjects, MDS remains a chronic disease and the primary goal is to achieve the best quality of life and delay progression into AML, which affects about one third of patients [6].

Therapy of MDS is based on the international prognostic scoring system (IPSS) risk stratification, which divides patients into four categories, who are then grouped into low risk (LR-MDS) and high-risk MDS (HR-MDS) [7]. More recently, a deeper dissection of cytogenetics abnormalities led to a revised IPSS (IPSS-R), which better stratifies low and high-risk MDS patients [8]. The treatment goals are different between the two groups, as well as the risk of subsequent evolution to AML. In the former, the treatment is mainly supportive (i.e., transfusion, erythropoiesis stimulating agents, ESAs, iron chelation) and aims at improving cytopenia symptoms and preventing possible complications (i.e., infections and bleeding). In the latter, the key point is delaying leukemic progression and improving survival.

Increasing data about recurrently mutated genes and chromosomal abnormalities are emerging due to the widespread availability of gene sequencing techniques, such as next generation sequencing (NGS) [2,9]. Efforts are being made to incorporate these biological data into the current disease classification and prognosis systems, to possibly refine IPSS risk stratification [10,11].

In this review, we provide an overview of the current and future strategies for the management of LR-MDS and HR-MDS patients. In particular, we focus on the unmet clinical needs that might be fulfilled through the targeting of novel biological pathways.

## 2. Diagnosis and Classification and Their Drawbacks

Diagnosis of MDS is based on the presence of one or more blood cytopenias (Hb < 10 g/dL, platelets < 100 × 10^9^/L, neutrophils < 1.8 × 10^9^/L), as a pre-requisite, plus one of the following: dysplastic features in ≥10% of nucleated cells in at least one lineage, <20% blasts in blood and bone marrow, >15% ring sideroblasts (RS) in bone marrow and characteristic cytogenetic or molecular findings [12]. Staining for iron with Prussian blue (Perls stain) is pivotal to detect MDS with RS (MDS-RS), and 80 to 90% of cases are associated with *SF3B1* mutation [13]. In the presence of such a mutation, the threshold of RS for diagnosis is reduced to ≤5%. Some chromosomal abnormalities are presumptive of MDS in cytopenic patients even in the absence of dysplasia, particularly chromosome 5q deletion. The latter identifies the del5q syndrome that carries peculiar clinical findings and deserves specific therapeutic management.

Bone marrow trephine biopsy is recommended, because of its prognostic potential regarding bone marrow cellularity and fibrosis. Hypocellular MDS, defined as <20% marrow cellularity, occurs in approximately 10% to 20% of cases and is associated with a higher response rate to immunosuppressive therapy and a favorable prognosis [14]. Marrow fibrosis (MF), on the other side, may predict an aggressive course with a significantly higher rate of progression [15].

The newly published 5th edition of the WHO classification introduces the term *myelodysplastic neoplasms* to replace myelodysplastic syndromes and emphasizes the dichotomy between MDS with “defining genetic abnormalities” versus “morphologically defined”. In the former, two distinct entities benefit from specific target therapies, namely MDS with low blasts and isolated 5q deletion (MDS-5q) and MDS with low blasts and *SF3B1* mutation (MDS-SF3B1, formerly MDS-RS). Furthermore, MDS with biallelic *TP53* inactivation (MDS-bi*TP53*) has been defined; it is identified by two or more *TP53* mutations, or one mutation with evidence of *TP53* copy number loss or loss of heterozygosity. Morphologically, hypoplastic MDS (MDS-h) and MDS with fibrosis (MDS-f) have been recognized as independent nosological entities [16].

Overall, the diagnosis of MDS still largely relies on the morphologic evaluation of marrow aspirate, which also depends on the skill level of the operator and may be confounded by several factors, including toxic agents, vitamin B12/folic acid deficiencies, and age itself. The evaluation of somatic mutations may aid the diagnosis in some settings, but may also be confounded by the presence of the “clonal hematopoiesis of indetermined potential” (CHIP) and by the “age related clonal hematopoiesis” (ARCH) [17,18] that are also found in the general elderly population. Therefore, cautious interpretation is warranted, also taking into account the variant allele frequency (VAF) as a surrogate for clonal size. The aforementioned prognostic scores (IPSS and IPSS-R) have been used to predict patients’ survival and evolution, and for treatment allocation. However, a large grey zone area remains for intermediate risk patients (i.e., those with intermediate-1 and -2 IPSS and intermediate IPSS-R), who may display heterogeneous outcomes that resemble more low or high-risk MDS. In this regard, the understanding of a series of somatic mutations mostly associated to worse outcome, such as TP53, ASXL1, NRAS, SFRS2 and EZH2 [19], led to recent attempts in refining the current prognostic models. An example is the recently published Molecular International Prognostic Scoring System (IPSS-M), which includes data regarding 38 gene mutations, possibly improving the risk stratification compared to the IPSS-R [11].

## 3. Unmet Clinical Needs of Current Therapies in Lower-Risk Patients

The current approach suggested for low-risk patients is mainly based on the improvement of cytopenias, especially anemia, through transfusions and ESAs. Several unmet needs emerge from the current treatment algorithm, including (a) patients that do not benefit from/not a candidate for ESAs, (b) del 5q patients that do not benefit from lenalidomide, (c) MDS-SF3B1 patients that do not benefit from luspatercept, (d) management of thrombocytopenic MDS, (e) drawbacks of chelating therapy, and (f) proper prognostic stratification of LR-MDS patients.

(a) Some patients may be not a candidate for ESAs, since a clear benefit is usually observed in those with a serum erythropoietin below 500 mU/mL and who are mildly transfused, less than twice a month [20]. Additionally, erythropoietin alfa at a standard dose of 40,000 units per week can induce erythroid responses (ER, per international working group, IWG-2006 criteria) in only about 1/3 of patients, and the benefit is transient [21]. This results in transfusion dependence, with consequent fluid and iron overload, and risk of alloimmunization. These patients are generally long-term survivors, but have a low quality of life, high level of medicalization, and frequent hospital admissions. On the other hand, disease burden leads to reduced survival probability, as compared to non-transfusion dependent patients [22]. The second line treatments that are currently used, but are not approved in most countries, include anti-thymocyte globulin (ATG) for MDS-h, hypomethylating agents (HMAs), and lenalidomide, which generally show efficacy in no more than 1/3 of cases. A recently proposed approach is the rechallenge of patients who lost response to the first ESA with an alternative agent after a wash out period [23]. This re-established erythroid response in up to 40% of subjects, mostly switching to originator epoetin alpha from other agents.

(b) MDS-5q is associated with lower rates and significantly shorter responses to ESA compared with other LR-MDS [24]. Thereafter, they are candidate for lenalidomide. In a placebo-controlled phase 3 trial, lenalidomide induced red blood cell transfusion independence (RBC-TI) in 27% of patients for >8 weeks and with a median duration of 31 weeks [25]. Limited efficacy is accompanied by possible toxicities, particularly hematologic ones that may require dose reduction and treatment discontinuation in this elderly frail population. Additionally, TP53 gene mutations are found in 20% of del(5q) patients and are associated with scarce responses to lenalidomide and a higher risk of AML progression [26]; the TP53 gene has been addressed as a possible target for novel therapies, as detailed in the following paragraphs.

(c) Similar to del5q patients, those with MDS-RS are commonly transfusion dependent and show low and short-term responses to ESAs. Lately, luspatercept has been shown to be beneficial in this subgroup. It is an activin receptor fusion protein that alters transforming growth factor (TGF-) beta signaling, thus reactivating late differentiation and maturation of erythroid progenitors [27]. In a randomized placebo-controlled phase III trial, 153 of 229 patients with very low-, low- and intermediate-risk MDS-RS received subcutaneous luspatercept every 3 weeks; 38% of patients achieved RBC-TI for 8 weeks or longer and 33% remained transfusion independent for more than 48 weeks [28]. Unfortunately, the majority of patients still require transfusions. Additionally, no significant differences were noted in AML evolution between the two treatment arms, confirming the unmet need for disease-modifying agents in this setting.

(d) No labeled drug is available for thrombocytopenic MDS. The watch and wait strategy is adopted in those with PLT > 30 × 10^9^/L and without active bleeding, and PLT transfusions are often ineffective due to alloimmunization. In patients that require therapy, the approach mainly consists of targeting the immune system to decrease PLT destruction and stimulating BM through TPO-RA and androgens. An attempt with steroids is often performed, at doses similar to those used in immune thrombocytopenia (ITP), particularly if anti-PLT autoantibodies are positive. The latter show, however, low sensitivity and specificity, and steroid response is usually limited and accompanied by harmful side effects. Aplastic anemia (AA)-like immunosuppressive treatment has also been studied in MDS; ATG +/− cyclosporine achieved a platelet response in 35–40% of cases of lower-risk MDS [29], with a major benefit in patients with hypoplastic bone marrow, HLA DR15, age < 60 years, trisomy 8 and low transfusion burden [30]. As in AA, horse ATG appears to achieve better results than rabbit ATG [29]. Regarding TPO-RA, in a randomized placebo-controlled phase II study, romiplostim significantly reduced the incidence of severe bleeding and platelet transfusions with the weekly 750-μg dose [31]. While there was a suspected increase in AML risk upon first analysis, this was not confirmed by the long-term follow-up [32]. Eltrombopag was tested versus a placebo in a randomized phase II trial, showing a platelet response in 47% of cases and reduction in bleeding events, with no safety concerns regarding AML evolution [33]. Finally, androgens can improve thrombocytopenia in one-third of cases, but the response is generally transient. Recent retrospective studies with danazol at a median dose of 400 mg/day [34] and stanozolol [35] in lower-risk MDS patients showed a variable hematologic improvement in 48–60%, again associated with hypocellular bone marrow. The most common side effects included masculinization and liver enzyme alterations, so baseline urologic and hepatic evaluation are advised. Overall, these strategies are effective in about 1/3 of cases, for a limited time span, and with potentially harmful side effects, thus advocating for novel treatment strategies in this group of patients.

(e) A further issue involves transfusion-dependent patients that are not candidates for or that show suboptimal responses to chelation therapy. Deferasirox at a dose of 14 mg/kg is useful in preventing the occurrence of cardiac or liver dysfunction, transformation to AML or death, and may prolong event-free survival of 1 year compared to non-chelated patients [36]. Nevertheless, its use requires caution in patients with altered renal function, and should not be considered if the glomerular filtration rate is <45 mL/min. The hepatic, ocular and cochlear toxicities demand prompt dose adjustments, with the consequent risk of suboptimal chelation. Since MDS patients are generally elderly and comorbid, optimal schedules of iron chelation are difficult to maintain. Non-nephrotoxic chelation strategies, such as deferoxamine, may be used but are difficult to manage, may require the presence of a caregiver (i.e., daily 8-12 h subcutaneous pump injections) and carry additional side effects [37].

(f) Finally, there is a need for better prognostication in LR-MDS. In fact, large series studies showed that IPSS and IPSS-R are unable to clearly differentiate the fate of LR patients, and the addition of molecular data into molecular IPSS does not seem to significantly change the picture [9]. However, studies that include NGS evaluation highlighted more favorable clinical features and response to ESAs in patients with <2 somatic mutations, and a better prognosis for the subjects that carried isolated SF3B1 mutations [9,38].

## 4. Unmet Clinical Needs and Current Therapies in Higher-Risk Patients

Therapeutic approaches in higher-risk MDS patients are aimed at changing the natural course of the disease, improving overall survival and reducing the risk of leukemic evolution. HMAs are preferred for cytoreduction both in transplant and non-transplant candidates, but only azacytidine (75 mg/m^2^ per day for 7 every 28 days) has been approved in Europe [39] so far. A randomized phase III trial that compared decitabine to the standard of care failed to demonstrate a survival advantage, possibly related to the high number of patients enrolled with poor-risk cytogenetics and treatment-related MDS [40].

Allo-HCT remains the only potentially curative treatment of higher-risk MDS patients, although it is restricted to a minority of patients due to advanced age. In this setting, cytoreductive therapy with HMAs or chemotherapy prior to HCT is advised for patients with ≥10% bone marrow blasts. Regarding conditioning, myeloablative approaches seem to reduce relapse risk and should, therefore, be proposed in young and fit patients [41].

The following unmet clinical needs are noteworthy: (a) some patients do not benefit from HMA, (b) some patients are not candidates for allo-HCT, (c) better understanding of the actual AML evolution risk, (d) identification of specific mutations harboring poor prognosis.

(a) HMAs can induce transient responses in ∼50% of treated patients, but are not able to eradicate the neoplastic clone. Ultimately, all patients will become resistant, including subjects that do not respond at all (primary resistance). These patients have a poor short-term prognosis, with scarce therapy options, mainly represented by best supportive care, ESAs, low-dose chemotherapy and lenalidomide [42]. Switching to an alternative HMA may be suggested in the case of poor tolerance, although few retrospective studies have showed a limited benefit; this may be due to the slightly different mechanisms of action of azacytidine and decitabine [43]. Additionally, although generally well tolerated, hematologic toxicity with prolonged cytopenias may occur after HMAs. This may be severe and difficult to manage in this population with intrinsic high infectious and bleeding risks. Finally, patients’ convenience with azacytidine is particularly low since it is administered for 7 consecutive days (or 5 + 2, to skip the weekend) at day care facilities.

(b) The second unmet need is the feasibility of allo-HCT that is available for a minority of fit patients only. HMA and AML-like chemotherapy may be a suitable option as a bridge to transplants. The advent of reduced intensity conditioning (RIC) regimens extended the indication to patients with comorbidities or reduced fitness. Additionally, the increasing use of unrelated or mismatched family donors further improved accessibility, and large cooperative group and single institution studies report 30 to 52 percent overall survival (OS) with these approaches [44,45]. Nevertheless, the vast majority of patients remains ineligible, stressing the need for newer disease modifying therapies for unfit patients.

(c) AML and MDS lie along a disease continuum, where little is known about the predisposing factors of leukemic evolution. An expanding field of study is the relationship between the marrow microenvironment and the positive selection of AML clones, likely linked to decreased immunosurveillance through decreased HLA antigen exposure on neoplastic cells and activation of immune checkpoints [46]. Among the latter, two major inhibitory molecules are involved in suppressing T-cell function, namely cytotoxic T lymphocyte-associated protein 4 (CTLA4) and programmed cell death protein 1 (PD-1) [47]. Furthermore, the role of checkpoint molecules of innate immunity, such as CD47 for monocyte/macrophage cells, is emerging and is the object of active investigation. Immune checkpoint inhibitors are already approved in other hematologic malignancies and their role in MDS will be analyzed in the following paragraphs.

(d) Over the last few years, a number of studies have highlighted the detection and clinical impact of multiple genetic lesions in MDS. Despite the high heterogeneity of such findings, mutations in RUNX1, TP53 or EZH2 genes were consistently associated with an adverse prognosis, while those of SF3B1 were linked to prolonged survival and favourable outcomes [2,48,49]. Two major studies [50,51] evaluated the prognostic value of somatic mutations after allogeneic stem cell transplantation and concluded that TP53, RUNX1, ASXL1, JAK2 and RAS pathways alterations were associated with significantly shorter overall survival. Even though the use of molecular findings to guide the treatment decision is still at its start, newer molecules, developed to overcome specific pathway alterations, will possibly change the picture.

## 5. New Approaches for Low-Risk MDS Patients

Novel therapies for LR-MDS (Table 1) include drugs that target epigenetics, molecular pathways involved in cell maturation, ineffective erythropoiesis, response to hypoxia, telomeres elongation, inhibitors of pro-inflammatory immune pathways, and inhibitors of splicing mutations (Figure 1).

### 5.1. Targeting Epigenetic Alterations

Although typically reserved for HR-MDS, azacitidine and decitabine have demonstrated efficacy also in several phase II studies designed for LR-MDS patients, although with only modest responses, represented by TI rates that range from 14.3% to 25% [52,53,54]. Oral formulations are now available, highly improving patients’ convenience, although with possible gastrointestinal symptoms. The efficacy of oral azacytidine has been evaluated in 216 LR-MDS patients in a randomized phase III placebo-controlled trial, where RBC-TI was achieved in 31%. However, early deaths occurred in the treatment arm, most related to infections in patients with significant pretreatment neutropenia, requiring further evaluation [55]. Oral decitabine/cedazuridine (cytidine deaminase inhibitor) led to RBC-TI in 13 out of 27 (48%) patients. The agent was generally well-tolerated and resulted in median leukemic-free survival and overall survival that exceeded 32 months [56]. Of note, these trials evaluated lower doses of both decitabine (20 mg/m^2^ × 3 instead of 5 days) and azacytidine (75 mg/m^2^ × 3–5 instead of 7 days). Late phase clinical trials with oral HMAs are ongoing in LR-MDS.

### 5.2. Targeting Intracellular Pathways

The TGF-β—SMAD2/SMAD3 pathway is overactive and contributes to anemia in MDS. Luspatercept is the only approved agent in this setting and is being studied also in non-del(5q) LR-MDS, in combination with lenalidomide (NCT04539236). Additionally, since a better response was noted in MDS-RS with a low transfusion burden, a trial comparing luspatercept with ESA first-line treatment is currently ongoing (NCT03682536). A similar agent, sotatercept, an activin receptor IIA (ActRIIA) ligand, led to a hematologic improvement—erythropoietic (HI-E) for 36 out of 74 (49%) heavily pretreated LR-MDS patients, with 27% achieving RBC-TI for a minimum of 56 days [57]. Both agents were administered subcutaneously every 21 days.

Oral galunisertib, a TGF-beta receptor I kinase/ALK5 inhibitor, achieved HI-E in a total of 10 of 41 patients (24%) [58]. Patients with HI-E had a pronounced stem cell differentiation arrest, as demonstrated by a higher baseline proportion of lineage negative (Lin−) stem cells or CD33−/CD34+/CD38−. These findings suggest that even early progenitor differentiation block could originate from increased TGF-beta signaling, and therefore could be responsive to ALK5 inhibition. This might pave the way to the development of pre-treatment laboratory markers to be implemented in clinical practice.

Hypoxia-inducible factor α (HIF-α) is a key transcription factor that mediates physiologic RBC production in response to hypoxic conditions and is normally degraded by HIF prolyl hydroxylase (HIF-PH). In the dose escalation portion of a phase III trial [59], the HIF-PH oral inhibitor roxadustat led to TI in 9 out of 24 (37.5%) patients and the recommended 2.5 mg/kg dose trice per week is currently being evaluated (NCT03263091).

Increased telomerase activity (TA) and abnormal enzymatic activity of human telomerase reverse transcriptase (hTERT) are believed to play an important role in the development of MDS [60]. Imetelstat is a telomerase inhibitor administered intravenously once a month that showed efficacy in a phase II trial. Transfusion independence was achieved in 37% and 23% of patients at 8 and 24 weeks, respectively. Responses were higher in (HMA)/lenalidomide naïve and del5q patients. In addition, bone marrow responses, reduction in RS and in the mutations’ VAF were observed, suggesting a disease-modifying activity [61].

Pexmetinib is a dual inhibitor of p38 and Tie2, administered orally every 28 days, until progression or intolerance. The role of the p38 MAPK family is to control cytokine biosynthesis and cellular response to hypoxia-related stress, while Tie2 role is poorly characterized. An abnormal activation and expression of both p38 MAPK and Tie2 have been recently implied in the pathophysiology of MDS, and a phase I trial with pexmetinib is ongoing. Preliminarily, 14 out of 44 (32%) patients achieved HI. Notably, 13 of them had been previously treated with an HMA, suggesting that both p38 and tie2 may be associated with HMA resistance [62].

### 5.3. Targeting Immunity

Self-reactive immune attack and pro-inflammatory pro-apoptotic cytokines are involved in the pathogenesis of cytopenia in MDS, especially in the hypoplastic subtype. The IL-1 pathway promotes a proinflammatory bone marrow microenvironment, facilitating the neoplastic clone [63]. The anti-IL1β antibody canakinumab is currently being evaluated for the treatment of LR-MDS in a phase II trial (NCT04239157). Innate immune sensors, such as Toll-like receptor 2 (TLR2), are overexpressed in bone marrow CD34+ cells from LR-MDS patients and gain of function mutations correlate with progression. Tomaralimab is a fully humanized monoclonal antibody that inhibits TLR2, as evaluated in a phase I/II study. Patients were treated intravenously every 4 weeks for up to 9 cycles, and 6 out of 22 (27%) heavily pretreated patients achieved TI for at least 2 cycles [64].

### 5.4. Targeting Specific Mutations

RNA splicing is an essential cellular function involved in hematopoiesis, DNA damage response and epigenetic modification. Mutations in SF3B1, SRSF2, U2AF1 and ZRSR2 can be identified in over 50% of MDS cases and may be suitable for on-target therapies.

H3B-8800 is an oral inhibitor of the SF3b complex, as evaluated in a phase 1 study that included patients with AML and MDS (including 20 HR and 21 LR). Only 15% reached TI for over 56 days. However, 5 out of 15 (33%) patients with a splicing mutation reached TI, suggesting a greater activity in this setting. The toxicity profile and dosing are still not fully understood, and a case of pancytopenia and marrow aplasia during the first week of study was observed during dose escalation [65].

Overall, all these novel drugs seem effective in about 1/3 of patients, mainly inducing erythroid improvement and transfusion independence in a subgroup. The data are still considered preliminary in the assessment of the duration of response and impact on leukemic evolution, and post-marketing evaluation in the real world will be fundamental. Quality of life is a key endpoint for this long-term surviving population, and most of these agents are likely to improve patients’ reported outcomes (PROs) and patient convenience, via delayed subcutaneous or intravenous administrations (i.e., luspatercept, sotatercept, imetelstat) and oral administration (oral HMAs, galunisertib, roxadustat, pexmetinib, and H3B-8800). The safety profiles appear to be reassuring, but more data on long-term follow ups are needed. Finally, no novel agents for thrombocytopenic MDS patients are under evaluation.

## 6. New Approaches for High-Risk MDS Patients

Novel therapies for HR-MDS (Table 2) can be cathegorized in four main classes according to the mechanism of action: drugs targeting epigenetic alterations, intracellular pathways, specific mutations and enhancer of immune response.

### 6.1. Targeting Epigenetic Alterations

Guadecitabine is a second generation HMA resistant to degradation by citidine deaminase, administered subcutaneously. A phase 1/2 included both treatment naïve and relapsed HR-MDS patients who were randomized to receive guadecitabine 60 or 90 mg/m^2^. A total of 51% of naïve and 43% of HMA refractory patients obtained a response. Based on these data, the recommended dose was 60 mg/m^2^ on a 5-day schedule [66].

In a randomized crossover phase III trial, oral decitabine/cedazuridine yielded similar systemic exposure, efficacy and safety in comparison with decitabine 20mg/m^2^ intravenously (IV), being FDA approved in 2020 for all subtypes of adult MDS [67].

Regarding combination therapies, azacytidine was studied in combination with pevonedistat. It is the first inhibitor of NEDD8-activating enzyme (NAE), which normally activates cullin-RING E3 ubiquitin ligases (CRLs). Inhibition of NAE by pevonedistat prevents degradation of CRL substrates that are integral to tumor cell growth, leading to cancer cell death. In a phase II study, patients were randomized 1:1 to receive either the combination or azacytidine alone. In HR-MDS, pevonedistat + azacytidine led to longer event-free survival (EFS, median 20.2 versus 14.8 months), complete remissions (CR, 51.7% versus 26.7%) and duration of response (DOR, median 34.6 versus 13.1 months). Pevonedistat is administered at a dose of 20 mg/m^2^ intravenously on days 1/3/5, in 28 days cycles. Of note, the combination appears less myelosuppressive than azacytidine and venetoclax; hence, it is more feasible as outpatient treatment [68]. Despite these positive aspects, in the recently published phase 3 trial, the azacytidine plus pevonedistat combination failed to meet the primary endpoint of longer EFS and no evident OS advantage was noted. However, improved OS was reported in patients that received more than three cycles, underlining the value of continuing therapy for more than three cycles [69].

Other effective partners of HMAs are venetoclax and checkpoint inhibitors, which are outlined in the following paragraphs. Finally, it is also worth mentioning several agents, such as lenalidomide, vorinostat, eltrombopag, vosaroxin, volasertib or histone deacetylase (HDAC) inhibitors, which have been studied in combination with azacytidine but did not improve response rates [70].

### 6.2. Targeting Intracellular Pathways

Inhibiting the antiapoptotic BCL2 protein is a well-known strategy in AML, where venetoclax showed clear efficacy in both relapsed [71] and de novo [72] cases in combination with HMAs. Several trials are now evaluating its role in HR-MDS patients, with encouraging preliminary data. In a phase Ib study that included treatment naïve HR-MDS patients, the combination venetoclax + azacytidine achieved an overall response (ORR) rate of 80%, including 40% CR and 40% marrow CR (mCR). For the 31 CR patients, the median duration of response was 13.8 months and median OS was 28.6 months [73]. As for AML, the major toxicities of this treatment strategy are tumor lysis syndrome, hematologic toxicity, with about 1/3 of subjects developing neutropenia and thrombocytopenia, and the occurrence of infectious complications. Similar findings have been recently reported in another phase 1 trial, with an ORR of 87%, and the establishment of the phase 2 dose as azacitidine 75 mg/m^2^ for 5 days, plus venetoclax 400 mg for 14 days [74].

Rigosertib is a RAS inhibitor that blocks the downstream signaling of multi-kinases, such as phosphoinositide 3-kinases (PI3Ks), thereby hindering cancer cell growth. It is administered via 72-h continuous intravenous infusion every other week and was evaluated versus the best supportive care (BSC) in a phase III trial, failing to demonstrate a survival benefit [75]. However, a subanalysis of the study showed efficacy in patients that were primarily resistant to HMA and in IPSS-R, very high-risk patients [76], leading to a subsequent ongoing trial in this setting (NCT02562443).

### 6.3. Targeting Immunity

Immunologic therapies for MDS include (a) therapies that target MDS-associated antigens and (b) those that target immune effector cells to elicit an antileukemic effect.

(a) The former includes antibody-drug conjugates and naked antibodies, mainly directed against CD33 and CD123 on neoplastic cells. The two antibody drug conjugates against CD33 and CD123, gemtuzumab ozogamicin [77] and talacotuzumab [78], have not yielded favorable results so far. Other options include the antibody drug conjugate tagraxofusp (NCT03113643) and the naked antibody BI 836858 (NCT0224070).

(b) The latter category includes immune checkpoint inhibitors, bispecific antibodies, and chimeric antigen receptor (CAR) T-cells.

The expression of PD1 and CTLA-4 is enhanced by HMA treatment and may be a possible mechanism of resistance [79]. In an ongoing phase II trial, the PD1 inhibitor nivolumab and the CTLA-4 inhibitor ipilimumab (intravenously, once and twice per 28 days cycle, respectively) were studied in combination with (HMA naïve cohort) or without (HMA failure cohort) azacytidine and an update on 26 patients was presented at ASH in 2020 [80]. In the HMA failure cohort, the ORR was 36% with two cases of CR (18%), suggesting only moderate activity in case of precedent HMA exposure. In the frontline cohort, the ORR was 67%, with five achieving CR (33%). Longer follow ups in larger cohorts are needed to further validate the efficacy of these regimens. Conversely, PDL1 blockade was less effective, as durvalumab [81] and atezolizumab [82] failed to demonstrate clinically meaningful efficacy. Moreover, atezolizumab was associated with a high rate of early death (21 out of 42 patients), leading to premature trial discontinuation.

T-cell immunoglobin domain and mucin domain-3 (TIM-3) is an immune checkpoint with a complex regulatory role in both adaptive and innate immunity. In a recent phase Ib trial, the anti-TIM-3 antibody sabatolimab, administered intravenously every 2 or 4 weeks, was combined with decitabine in HR-MDS patients. The preliminary results are encouraging, with 8 out of 16 (50%) patients achieving CR after a median of 2 cycles [83].

Recently, CD47, a macrophage immune checkpoint, is gaining attention. This molecule inhibits phagocytosis and allows immune evasion by leukemic cells. Magrolimab is an anti-CD47 antibody administered intravenously every 1 or 2 weeks, evaluated in a phase Ib study in combination with azacytidine. The CR and objective response (OR) rates were 33% and 75%; the 12- and 24-month overall survival (OS) rates were 75% and 52%, respectively; median OS was not reached after 17.1 months of follow-up. Importantly, this agent maintained a great efficacy also in TP53 mutated cases [84]. The most common grade 3/4 toxicities included anemia (47%), neutropenia (46%) and thrombocytopenia (46%). In particular, anemia is also due to CD47 expression in erythrocytes. This hematologic toxicity led to a temporary pause in the phase 2 study, and updated results are awaited.

Bispecific antibodies are designed to target a tumor-specific antigen and a T cell activating antigen, typically CD3. Flotetuzumab is an anti CD3/CD123 bispecific antibody that was evaluated in a phase I study on 45 relapsed/refractory (R/R) AML and MDS patients (89% AML and 11% MDS); 14 completed at least 1 cycle and 6 (43%) reached an ORR, as per IWG criteria (three CR, one CR with incomplete hematologic recovery, one morphologic leukemia free state (MLF), one partial remission) [85].

A similar agent that targets CD3/CD123 and APVO436 is being evaluated in a phase I dose escalating trial, and preliminary results on 22 AML and 6 MDS patients are available. A weekly dose up to 18 µg has been evaluated and a blast reduction was achieved in two patients. Of note, one MDS patient maintained a durable response for eleven cycles before progressing [86].

Finally, following the benefit in lymphoid malignancies, several trials are investigating the role of CAR-T cell therapy in myeloid neoplasms. CAR-T targets on MDS cells need a high degree of specificity to avoid exerting toxicity on healthy stem cells and myeloablation [87]. In AML, several targets, such as CD33, CD123 and NKG2D, are under investigation [88]. In MDS, a study that evaluated NKG2D CAR-T cells in 9 R/R AML and MDS patients reported evidence of stable disease after CAR T-cell administration [89]. Additionally, several CAR-T constructs, including combinatorial targets, are being evaluated, including CD123-CD33 (NCT04156256), CLL1-CD33 (NCT03795779), or CD33-IL15 (NCT03927261).

### 6.4. Targeting Specific Mutations

Mutations in isocitrate dehydrogenase-2 (IDH2) and IDH 1 occur in around 5% and 3% of patients with MDS, respectively. Activity of mutant IDH proteins results in hypermethylation of DNA and histones, blocking hematopoietic differentiation. Early phase studies have investigated both IDH1 inhibitor ivosidenib [90] and IDH2 inhibitor enasidenib [91] monotherapy in HR-MDS, with nearly 50% of patients obtaining CR. Adverse events of special interest with these drugs include tumor lysis and differentiation syndromes and require proper monitoring.

Strategies that target TP53-mutated MDS are also being explored. Eprenetapopt (APR-246) is a small-molecule, which causes thermodynamic stabilization of the p53 protein by shifting the equilibrium toward the wild-type conformation, thus restoring protein function. In a phase 1b/2 trial, the combination of eprenetapopt and azacitidine was well tolerated and elicited a response (ORR) in 62% of patients [92]. The combination was subsequently investigated in a phase 3 trial, in comparison with azacytidine alone (NCT03745716), but failed to meet the primary endpoint of complete response attainment.

Overall, new generation HMAs will likely improve patient convenience, even in the HR-MDS setting. Novel drugs have interesting mechanisms of action, many of which are immunoregulatory. These drugs are struggling to improve the outcomes obtained with HMAs, and further investigation is needed to evaluate the best positioning and combinations. The activity of some agents as monotherapy in patients who did not benefit from HMAs carries the promise to prolong response and survival in these patients with particularly dismal outcomes.

## 7. Conclusions

Current treatment options in MDS are selected according to the individual risk stratification, which is based on IPSS and IPSS-R scores. However, the boundaries between low and high-risk MDS are the object of continuous refinement, as demonstrated by the recent inclusion of molecular features in the IPSS-M. These changes will impact on treatment allocation and will possibly aid to clarify the existing “grey zone” of intermediate risk patients.

In LR-MDS patients, the goal is to improve cytopenias and to preserve an acceptable quality of life, often affected by the transfusion burden and frequent hospital admissions. Patients’ convenience may be improved by oral and subcutaneous formulations that also lower medicalization and hospital attendance. Promising agents for LR patients mostly target the intracellular pathways involved in cell maturation and apoptosis. The HIF-PH oral inhibitor roxadustat induced transfusion independence in 37.5% of patients and similar results were obtained with imetelstat. The latter may also exert disease modifying activity, as a reduction in BM ring sideroblasts and the mutations’ VAF was observed. Targeting epigenetic alterations with HMA also showed efficacy, with decitabine/cedazuridine reaching encouraging rates of long-lasting transfusion independence. Further investigations are needed to understand the best positioning of these novel agents in the treatment algorithm of LR-MDS, and to explore possible combinations. For instance, it might be possible that targeting both early and late-stage erythropoiesis by combining luspatercept and ESA may be beneficial, as demonstrated in murine models [93]. Finally, some neglected groups, such as hypocellular, fibrotic, and thrombocytopenic MDS, are poorly addressed by newer agents. Since the recent 5th WHO classification recognizes these forms as independent nosologic entities, their representation in future trials should be implemented.

Regarding HR patients, altering the natural history that prevents AML evolution is of utmost importance. However, allo-HCT is the only curative option so far to be performed according to age and fitness.

Several agents are under study in this setting, usually in combination with a HMA backbone (as for venetoclax and checkpoint inhibitors). Nivolumab and ipilimumab reached encouraging results in the treatment-naïve setting, in addition to magrolimab, confirming the critical role of immune escape in advanced disease. Of note, the latter maintained its efficacy in the presence of TP53 mutation, warranting further investigation in this poor risk subset. Although a combination approach is supposed to increase the rate of response through additive or synergistic effects, none of these investigational agents combined to HMA seemed to drastically improve the outcome as compared to HMA alone, nor to clearly revert disease course. The implementation of new generation immunotherapies, such as bispeciphic antibodies and CAR-T cells, will possibly have a favorable impact in the near future.

Finally, the heterogeneity and frailty of MDS patients remain an obstacle to face. This suggests that we must focus on the patient’s situation as a “whole”, including their comorbidities, expectations, and quality of life. The choice and combination of old and new drugs may have the potential to meet this need.

## Figures and Tables

**Figure 1 cancers-14-04941-f001:**
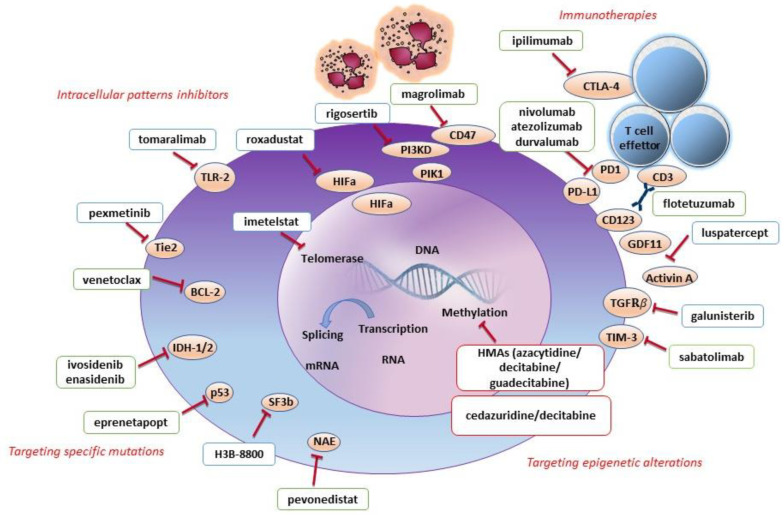
Novel therapies for myelodysplastic syndromes: immunologic, epigenetic and molecular targets. The physiopathology of myelodysplasia implies several pathogenic mechanisms, including epigenetic alterations, the disruption of intracellular molecular pathways, and the derangement of immunosurveillance that favors leukemic immune escape. All these pathogenic mechanisms may represent a target for novel drugs. For instance, immune checkpoint inhibitors, such as nivolumab, atezolizumab, durvalumab, ipilimumab, magrolimab, are able to restore T cells and adaptive and innate immunity to target dysplastic hematopoietic stem cells. Other immunotherapies include bispecific antibodies, such as flotetuzumab, and in the future, CART cells. Drugs that target intracellular pathways include luspatercept, galunisertib, sabatolimab, pevonedistat, imetelstat, pexmetinib, venetoclax, tomaralimab, roxadustat, and rigosertib and these act on various levels in cell maturation, survival/apoptosis, ineffective erythropoiesis, response to hypoxia, and telomeres elongation. Targeting epigenetic alteration by hypomethylating agents such as azacytidine, decitabine and novel agents (guadecitabine, decitabine/cedazuridine) restores MDS maturation. Finally, specific mutation inhibitors include H3B-8800, ivosidenib, enasidenib and eprenetapopt, which restore hematopoietic differentiation. Novel drugs for low-risk MDS are represented in blue boxes, drugs for high-risk MDS in green boxes and drugs for low-risk and high-risk MDS in red boxes.

**Table 1 cancers-14-04941-t001:** New drugs for lower risk patients.

New Active Principle	Mechanism of Action	Efficacy in Evaluable Patients	Clinical Trial
Imetelstat	Telomerase inhibitor	N = 38 R/R to ESAs 8-week TI rate of 37% for a median duration of 20 months	NCT02598661 phase II/III
Pexmetinib	p38/Tie2 inhibitor	N = 44 HI rate of 32%	NCT00916227 phase I
Galunisertib	TGF-β receptor type 1 kinase (ALK5) oral inhibitor	N = 41 HI rate of 24.4%	NCT02008318 phase II
Oral azacytidine (CC-486)	HMA DNA/RNA methyltransferases inhibitor	N = 216 RBC-TI rate of 31% with a median duration of 11.1 months	NCT01566695 phase III
Tomaralimab (OPN-305)	TLR-2 inhibitor	N = 22 TI for at least 2 cycles in 27% (major responders)	NCT02363491 phase I/II
Roxadustat	HIF-PH inhibitor	N = 24 TI rate of 37.5% for ≥56 consecutive days within the first 28 weeks	NCT03263091 phase III
Cedazuridine/decitabine (ASTX727)	Fixed-dose combination of the HMA decitabine and the novel cytidine deaminase inhibitor cedazuridine	N = 27 RBC-TI rate of 48%	NCT03502668 phase I/II

R/R, relapsed/refractory; ESAs, erythropoiesis stimulating agents; TI, transfusion independence; HI, hematologic improvement; TGF-β, transforming growth factor-beta; RBC, red blood cell; TLR-2, Toll-like receptor 2; HIF-PH, hypoxia-inducible factor prolyl hydroxylase; HMA, hypomethylating agent; DNA, deoxyribonucleic acid; RNA, ribonucleic acid.

**Table 2 cancers-14-04941-t002:** New drugs for high-risk patients.

New Active Principle	Mechanism of Action	Efficacy in Evaluable Patients	Clinical Trial
Guadecitabine	HMA, inhibits DNA/RNA methyltransferases	N = 105 ORR 51% treatment naïve patients ORR 43% R/R patients	NCT01261312 phase I/II
Pevonedistat + azacytidine	NAE first inhibitor	N = 58 CR rate of 51% with a median duration of response of 34 months	NCT02610777 phase II
Venetoclax + azacytidine	BCL-2 inhibitor	N = 78 CR rate of 40%	NCT02942290 phase Ib
Nivolumab + ipilimumab +/− azacytidine	Anti PD1 and anti CTLA4 immune checkpoint inhibitors	N = 26 CR rate of 18% in HMA failure cohort (N = 11) CR rate of 33% in HMA naïve cohort (N = 15)	NCT02530463 phase II
Sabatolimab + decitabine	Humanized anti-TIM-3 antibody	N = 16 CR rate of 50%	NCT03066648 phase I
Magrolimab + azacytidine	Anti CD47 immune checkpoint inhibitor	N = 33 ORR 91%, CR rate of 42%	NCT03248479 phase I
Flotetuzumab	CD123 X CD3 bispecific antibody	N = 14 Patients with either R/R AML or MDS ORR 43%	NCT02152956 phase I
Ivosidenib	mutant IDH1 inhibitor	N = 26 ORR 69%, CR rate of 46%	NCT03503409 phase II
Enasidenib	Mutant IDH2 inhibitor	N = 17 ORR 53%	NCT01915498 phase I

HMA, hypomethylating agent; DNA, deoxyribonucleic acid; RNA, ribonucleic acid; NAE, NEDD8-activating enzyme; ORR, overall response rate; CR, complete remission; TIM-3, T-cell immunoglobulin domain and mucin domain-3; MDS, myelodysplastic syndrome; IDH1, isocitrate dehydrogenase 1; IDH2, isocitrate dehydrogenase 2; r/r, relapsed/refractory; AML, acute myeloid leukemia.

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
