# Peer review of "Novel Therapies for Unmet Clinical Needs in Myelodysplastic Syndromes"

_cancers, 2022, doi:10.3390/cancers14194941_

Round 1

Reviewer 1 Report

Interesting review paper including all the relevant topics. Corrections are added in the file. 

Author Response

We would like to thank the Referee for revising our article and performed the suggested changes.

Reviewer 2 Report

This is a nice and comprehensive review on novel treatments in MDS. The organization of the manuscript is original and makes the paper easily readable.

I would suggest to the Authors to stress more the pitfalls of each novel agent : i.e. the unsatisfactory result of AZA-pevo combination (EFS primary endpont not met in phase III study and no evident OS advantage in AZA-Pevo arm) or the toxicity of magrolimab that led to a  suspension of the phase II study.

Author Response

We thank the Referee for revising our article and for the positive feedback. We appreciated the suggestions and added some sentences to stress the pitfalls of certain novel agents:

Pevonedistat: “Despite these positive aspects, in the recently published phase 3 trial the azacytidine plus pevonedistat combination failed to meet the primary endpoint of longer EFS and no evident OS advantage was noted. However, improved OS was reported in patients receiving more than 3 cycles, underlining the value of continuing therapy for >3 cycles [PMID: 35728048].”

Venetoclax plus HMA: “. As for AML, the major toxicities of this treatment strategy are tumor lysis syndrome, hematologic toxicity, with about 1/3 of subjects developing neutropenia and thrombo-cytopenia, and the occurrence of infectious complications.”

Magrolimab: “Magrolimab is an anti-CD47 antibody administered intravenously every 1 or 2 weeks, evaluated in a phase Ib study in combination with azacytidine. CR and objective re-sponse (OR) rates were 33% and 75%; 12- and 24-month overall survival (OS) rates were 75% and 52%, respectively; median OS was not reached with 17.1 months of fol-low-up. Importantly, this agent maintained a great efficacy also in TP53 mutated cas-es82. The most common grade 3/4 toxicities included anemia (47%), neutropenia (46%), thrombocytopenia (46%). In particular, anemia is also due to CD47 expression on erythrocytes. This hematologic toxicity led to temporary hold of the phase 2 study, and updated results are awaited.”

IDH1/2 ihibitors:” Adverse events of special interest with these drugs include tumor lysis and differentiation syndromes, and require proper monitoring.”

Reviewer 3 Report

The review is well described.

Please, add the paper for description of azacitidine and venetoclax (Lancet Haematol 2022, in press:

Azacitidine plus venetoclax in patients with high-risk myelodysplastic syndromes or chronic myelomonocytic leukaemia: phase 1 results of a single-centre, dose-escalation, dose-expansion, phase 1–2 study

  • Alexandre Bazinet,
  • Faezeh Darbaniyan,
  • Elias Jabbour,
  • Guillermo Montalban-Bravo,
  • Maro Ohanian,
  • Kelly Chien,
  • Tapan Kadia,
  • Koichi Takahashi,
  • Lucia Masarova,
  • Nicholas Short,
  • Yesid Alvarado,
  • Musa Yilmaz,
  • Farhad Ravandi,
  • Michael Andreeff,
  • Rashmi Kanagal-Shamanna,
  • Irene Ganan-Gomez,
  • Simona Colla,
  • Wei Qiao,
  • Xuelin Huang,
  • Deborah McCue,
  • Bailey Mirabella,
  • Hagop Kantarjian,
  • Guillermo Garcia-Manero. ).
  •  

Author Response

We thank the Referee for revising our manuscript and for the helpful suggestion. A sentence and the relative reference have been added:

“Similar findings have been recently reported in another phase 1 trial, with an ORR of 87%, and the establishment of the phase 2 dose as azacitidine 75 mg/m2 for 5 days plus venetoclax 400 mg for 14 days [PMID: 36063832].”
